# Scale Estimation for Design Decisions in Virtual Environments: Understanding the Impact of User Characteristics on Spatial Perception in Immersive Virtual Reality Systems

Sahand Azarby * and Arthur Rice

College of Design, North Carolina State University, Raleigh, NC 27607, USA

* Correspondence: sazarby@ncsu.edu

**Abstract:** User spatial perception in different virtual environments may vary based on specific user characteristics and the features of the Virtual Reality (VR) system. This research explored the impacts of user characteristics such as age, gender, and design knowledge on spatial decision-making by comparing an Immersive Virtual Reality Interactive Environment (IVRIE) with a traditional Virtual Reality system (also known as desktop-based Virtual Reality system, abbreviated herein as the DT system). Users' spatial perceptions when using IVRIE and a DT system were studied with regard to the features of the different systems, including the types of immersion and interaction, users' perceptions of human body scale, and how the environments were explored. The factors affecting the two systems included texture variation, type of enclosure, and spatial function. Inferential testing using quantitative data was applied to identify differences between the two systems in terms of participants' actual design outcomes. The results showed that based on the type, spatial characteristics, and texture of spaces, perception filters could have both active and inactive roles in impacting the spatial decision-making of participants between the two systems. In addition, between the two systems, participant characteristics had more impact on size variations for both types of spaces—fully enclosed and corridors—for accommodating larger groups.

**Keywords:** immersive virtual reality; user spatial perception; spatial decision-making; spatial design; spatial function; architectural design; perception filter



## 1. Introduction

This study explored the impacts of specific characteristics of user spatial perception/cognition and spatial decision-making in VR systems and evaluated the potential functionality of these systems for architectural design education and practice. The overall objective was to examine how the combination of user background and system features impacts users' spatial perception in each system, an Immersive Virtual Reality Interactive Environment (IVRIE) and a desktop-based VR (DT system).

The objectives of this study were to specify the role of users' characteristics in shaping spatial perception and cognition in IVRIE and DT systems and identify possible differences in spatial decision-making between the two systems that resulted from a combination of user characteristics and system features.

The assumption was that some users' characteristics, including age, gender, former design experience, familiarity with 3D immersive VR, and level of experience using IVRIE in design, would affect decisions related to spatial factors. The research questions were as follows:

- Does using IVRIE (independent var. 1/condition1) affect users' spatial decisions related to scale/volume when developing a design for a designated spatial functionality (dependent var.) by changing their spatial perception/cognition (intervening var.), as compared with a DT system (independent var. 2/condition 2) and relative to users' particular properties of perception and performance in the virtual environment (covariate)?

- If the degree of difference between the two systems with regard to users' spatial decisions is significant, which particular properties of perception play the most prominent role in affecting those variations?
- If users' spatial decisions based on task-based design guidelines result in significant space size variations between the two systems, is there any connection between users' particular properties of spatial perception and the complexity of the designated spatial functionality of the space?

The hypotheses for this research study were as follows. First, some users' particular properties and characteristics would affect their spatial perception in the IVRIE and DT systems and result in differences in spatial decision-making. Second, differences in the level of presence and spatial perception between the IVRIE and DT systems would affect users' task-based design performance and spatial decision-making, resulting in different spatial design results between the two systems.

This paper comprises five sections: Research Background, Methodology, Results, Discussion, and Conclusion. The research background summarizes the literature review and previous studies focusing on VR systems. The next section describes the chosen methodology and research design, including the experiment design and applied methods for data collection and analysis. The Results section presents the analyses and findings of this study. In the Discussion and Conclusion sections, the summary of the study's overall findings and future vision are presented.

## 2. Research Background

Digital representations of architectural design have evolved from two-dimensional (2D) representations to high-quality immersive visualizations that can be explored by designers through dynamic spatial movements and immersion along various spatial scales and from a variety of viewpoints [1,2]. In architectural design, three-dimensional (3D) visualization in Virtual Reality (VR) has most often been used for the conceptualization of spatial factors such as the volume, depth, form, proportions, spatial relationships, and arrangement of virtual spaces [3]. VR-based 3D visualization and specific features related to transferring spatial data can facilitate users' understanding of architectural design concepts and positively impact design thinking, spatial perception, and user performance, for individuals ranging from experts to users with no design knowledge [4–6]. Previous studies have indicated that among the various available environments and media for design learning, VR has the potential to offer a wide range of possibilities for engaging design learners in digital design thinking processes and enhancing their creativity and problem-solving capabilities [2,7–11].

Visualization of a 3D design can be challenging for less experienced designers and anyone lacking knowledge of architectural design when they view a 2D design. Immersive visualization in VR can facilitate the understanding of various spatial design factors such as scale, volume, dimensions, and relationships. VR as a learning environment/tool can positively impact a learner's understanding of critical design elements and accelerate the development of their spatial awareness and ability to rapidly prototype design ideas [12–15]. Thus, the ability of VR to provide full-scale 3D spatial data and facilitate the perception of data through immersion and interaction will lead to more accurate qualitative design representations than would other digital media [2,16–19].

Existing research has demonstrated VR to be a potential learning environment for three crucial aspects of design education. First, the efficiency of VR in providing high-quality visualization through a well-designed graphical user interface makes it more usable for various target groups. Second, the overall characteristics of VR utilization, including ease of use, enjoyment, and understandability, result in users developing a better understanding of complicated topics. Third, the specific features of VR, such as being able to walk and move, directly interact with design objects, gain access to permanent 360° viewsheds, and gather immediate feedback, offer more freedom and opportunities to users to test design ideas and enhance active learning experiences [7,20–22].

Most VR-related research exploring applications in architectural design education has highlighted context immersion, interaction, spatial presence, and perception as four major features critical for design learning in VR systems. In these user-centered studies, immersion, interaction, spatial presence, and spatial perception in virtual environments are emphasized as critical features that can enhance the design learning context [5,9,14,23]. The overall conclusion regarding the role of these features in enhancing the process of design learning is that "the combination of immersion and interaction in VR environments constructs the spatial presence of the user within the virtual environment, and spatial presence empowers spatial perception of spatial factors of design" [20,24]. Presence (specifically spatial presence) in Immersive Virtual Reality (IVR) results from the integration of users' various senses, such as attention, immersion, awareness, and degree of control. The "spatial presence of a user is based on the experience of feeling spatially located in a digital environment and involved in the process of perceiving/cognition of spatial data and becoming aware of the spatial characteristics of the virtual environment" [19,25,26].

IVR is a general name for VR systems and defined as "virtual environments in which the spatial data can be transferred to the user through experimenting [with] immersion and interaction while a user has an intuitive feeling of observing surroundings, interacting with design objects and receiving spatial data in human scale" [9,16,17].

"Spatial presence is linked to the combination of environmental factors and users' abilities/tendencies in observing the surroundings for shaping spatial perception within immersive environments" [19]. The perception of spatial factors in a virtual environment can be affected by user characteristics such as their spatial ability and the way they engage in spatial thinking to create a mental model of spatial relationships within a virtual system [16,27]. "Users' personal differences, characteristics, and imagery ability can affect experiencing . . . presence when using different virtual systems" [24,28].

IVR, which enables a user to have an active experience by sensing their immersion, is a unique type of interaction that distinguishes the technology from other well-known human/machine interactions utilized by users in other digital design media. "Indeed, VR enhances the type of interaction between a user and system from [an] external interaction to [an] internal one" [9,29,30].

Traditional 3D approaches in architectural design education and training have relied on using a mouse or keyboard as interaction interfaces, along with digitally generated structural forms and virtual spaces. These kinds of interactions are categorized as external and have been used for decades when designing in non-immersive virtual environments. "Interactions in IVR are considered internal because of the integration with immersion". These interactions (such as pulling, pushing, and grabbing) that occur between a user and virtual objects can be visualized immediately, and users can be informed of their interaction results in real time [21,29–31]. "The internal interactions can reach the level where users lose the awareness of interacting with a machine, and the interface disappears in their eyes when the sense of immersion changes from semi-immersive to fully immersive" [20].

When immersed in a VR environment, users experience by perceiving their existence and body within the virtual environment, feeling themselves to be a part of it and capable of interacting with it and the objects existing within it. Based on the power of their immersion in constructing users' spatial presence in the virtual environment, VR systems have been categorized as either semi-immersive or fully immersive [26,32,33]. In a semi-immersive VR environment, the user is partially immersed in a virtual world, with the sense of having indirect interactions with the virtual environment and existing objects within it. In contrast, in fully immersive environments, users are active observers with the perceived capability of having direct interactions with the environment, and feeling as if they are a part of that environment [24,34].

A user's perception level of spatial factors and relationships and their sense of involvement with the virtual environment may be affected by variations in their immersion and interactions sensed within the environment. In a fully immersive virtual environment, users feel themselves to be active observers within the environment, with the ability to

interact directly and intuitively with virtual objects existing within their surroundings. In semi-immersive virtual environments (also known as monitor-based VR or desktop-based systems (DT systems)), users are partially immersed in the virtual world, and the sense of indirect or remote interaction with virtual objects is dominant over more direct contact [17,24,32,34–36].

The levels of functionality and usability of immersive and semi-immersive environments in learning design foundations and concepts may be different. The results of recent studies have shown that although both types of VR systems can be used in a design educational context, the effective mechanisms for guiding, evaluating, and critiquing in the service of teaching/learning purposes can be different [2,4,9,16,37].

Some studies comparing users' level of presence in immersive and semi-immersive VR environments have found that the capability of a fully immersive environment to provide greater visual immersion for design learners increased their spatial presence and improved their performance, although none of these variables had functional effects on users who already had design knowledge and higher levels of expertise [32,38]. Additionally, it was concluded that, in DT systems, "higher users' spatial ability may positively impact their spatial cognition and result in easier interpretation of spatial relations between and within spaces in virtual models" [26,29].

Although research on VR system utilization in design pedagogy and practice is increasing, comparative and quantitative studies focusing on user-centered factors in design concept perception and performance are scarce [2,19]. Moreover, studies employing objective quantitative data to compare the functionality of different VR systems in terms of transferring spatial data and evaluating a combination of user characteristics and system features are rarer [10,33]. Most research exploring the performance of users in virtual environments and testing the usability of VR systems has utilized qualitative data extracted from users' self-evaluations of system functionality and usefulness [39,40]. Research identifying the power, differences, and strengths of VR systems in shaping user experience and enhancing spatial design should not rely solely on opinion-based data. This body of research also needs precise quantitative data extracted from measurable design results.

Although the features of VR systems (including the types of immersion, forms of interaction, and levels of spatial presence/perception) are fundamentally different, both fully and semi-immersive virtual environments are being used in architectural design education and practice. Identifying the ability of VR systems to maximize the perception of spatial factors through the lens of user experience will clarify the level of suitability and usefulness of these systems.

The main question is whether there are significant differences between fully immersive interactive and semi-immersive VR systems in transferring spatial data to users. Are there any specific characteristics of users' backgrounds which are related to possible differences between these systems in terms of spatial perception and decision-makings?

## 3. Methodology and Research Design

The methodological framework of this study was based on quantitative method research, and the procedure of conducting the experiments and data collection applies to quantitative, comparative, and within-subject experiment design phases; for data analyses, descriptive and inferential statistical testing were used [41,42]. Data collection relied on gathering two separate branches of quantitative data. One branch was the measurement of design results of spaces produced by participants using IVRIE and DT systems. These data were used to compare the differences in volume and area of designed spaces between the two systems. The other branch of quantitative data was extracted from the Participant Profile Questionnaire (PPQ). The proposed method for producing and collecting quantitative data in the first branch was the result of the authors gathering reliable and precise data extracted from real and measurable design results. The method utilized for collecting data by PPQ was based on proposed models and examples of 'background/experiment' questionnaires from previous research [43]. The collected data from the PPQ included

participant backgrounds in the format of qualitative data. These qualitative data were coded on numerical scales for application in statistical testing, along with quantitative data based on measurements of participants' design results in both systems. Participants completed the experiment in two steps. First, they redesigned a few 3D virtual spaces by utilizing a specific spatial/experiential guideline for each space, once in the DT system and then in IVRIE. After completing the design tasks in both systems, they answered the questions in the PPQ. The spatial/experiential guidelines consisted of four different guidelines, each specified for redesigning a space. Each guideline comprised information regarding a space's spatial function and capacity. Participants used the guidelines to spatially redesign the spaces using the DT system. The same process was repeated, and participants redesigned the same spaces using the same guidelines in the IVRIE.

The experiment design of this study was shaped based on a conceptual model concerning the hypotheses and research questions proposed for this study. In the experiment design conceptual model, two VR systems were used to compare the capability and possible functionality of these systems in transferring spatial data to the users with a wide range of particular property differences. In the proposed conceptual model, DT and IVRIE systems and their different features were identified as independent variables that could impact the design outcomes of systems' users as dependent variables through a net of mediator and moderator variables. This study's mediator factors (a mediator is a way in which an independent variable impacts a dependent variable [42,44]) were categorized into experiential/spatial factors and human/machine interaction factors. Experiential/spatial factors included spatial presence, spatial perception, spatial awareness, and spatial thinking. The factors were hypothesized to be different when using the DT system versus IVRIE because of the different features of the systems. The human/machine interaction factors included ease of use, enjoyment, control/active learning, and motivation. These factors are assumed to be different between DT and IVRIE because of differences in physical interaction interfaces and perceptual interactions (internal versus external interactions). The combination of these two mediator variables categories, "experiential/spatial" factors and "human/machine interaction" factors either directly or indirectly impact the way a user understands a system, perceives primary spatial data within the system, thinks about new spatial arrangements, and finally, completes design tasks by making spatial decisions based on achieved spatial cognition. "Spatial cognition/thinking" factors as intervention variables in the conceptual model consisted of all users' activities in each virtual environment from perception to performance, including spatial decision-making, spatial performance, and space spatial functionality definitions and interpretations. In the last segment of the conceptual model, the combinations' output of the mediator and intervening variables encountered the user's perception filters as moderator variables (also called covariates). Figure 1 presents the flowchart of this study's proposed conceptual model of experiment design.

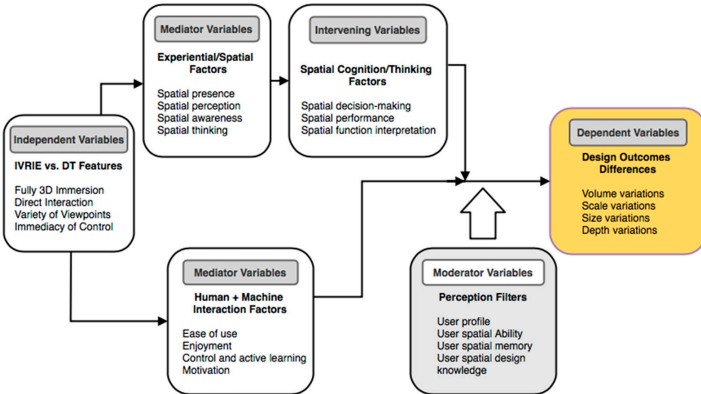

**Figure 1.** Proposed conceptual model for the experiment design.

User perception factors are the factors that can impact the spatial perception of users concerning their particular properties and characteristics, including spatial ability, spatial memory, and spatial design knowledge. Ultimately, the differences in design outcomes produced by each participant using both systems as dependent variables revealed the differences between these two virtual systems in conveying spatial data to the user for spatial decision-making and performance.

Based on the overall sequence and inclusion of various variables and determining factors in the conceptual model, the experiments in this study were designed and conducted in five steps. The first three steps consisted of the preparation process of the experiments, and in the second phase, the processes of performing the experiment and data collection were completed.

In step 1, systems and software were selected and set up regarding the differences between an interactive semi-immersive virtual environment (DT system) and a fully interactive immersive environment (IVRIE system) required for this study. SketchUp® software was selected for the DT system, and the VR Sketch® program was chosen as IVRIE. This study used a conventional workstation for the DT system, including a high-performance computer and a 27-inch full HD monitor, keyboard, and mouse as interaction interfaces. The VR Sketch® program in IVRIE was provided through an Oculus Rift device set, including two headset sensors, a headset, and two controllers serving as interaction devices, connected to the same high-performance computer. Participants used the DT system while sitting at a desk. They worked in IVRIE by putting on the Oculus headset and moving around in the designated area for the experiment in a design lab room. In step 2, virtual models were developed using SketchUp® software and spatial/experiential guidelines were completed. The process of matching the spatial function of spaces with designated spatial/experiential guidelines was finalized through long discussion, review, and testing sessions.

Participants worked and manipulated the developed models directly within SketchUp® software in the DT system. By exporting the models to the VR Sketch® extension, the models became available as immersive interactive 3D spaces for the IVRIE section of the experiment. The link between the SketchUp files and the VR Sketch plug-in was immediate. The researchers could see the models on the screen simultaneously when participants were working on them using the headset and controllers. The criteria and characteristics of models and their designated spatial/experiential guidelines were selected based on the quantitative data required to be extracted from measuring the volume/area or scale of the spaces designed by participants in both systems. Each participant worked on two sets of models in each system. Each set (in this study, we called them scenario) had four spaces, including two corridors and two enclosure spaces. The difference between the sets of models (scenarios) was the type of texture, in which all the available spaces in one scenario did not have any texture (plain). In contrast, all four spaces had a brick-shaped texture in the other set. The reason for providing two sets of models in each package with different texture presentations and including four spaces in each set with two types of enclosures was to obtain comprehensive data by integrating the roles of texture (plain vs. patterned), enclosure types (open-ended corridor vs. fully enclosed space), spatial functions (walking vs. gathering), and spatial capacity (number of people within each space) in the spatial decision-making of participants using each system. Four spatial/experiential guidelines were used: two designated for corridor spaces and the other two for enclosure spaces. Each guideline was used twice by the participants, first for redesigning a space with untextured walls and then for a pair of the textured spaces. The same procedures were followed for both the DT and the IVRIE systems. Each guideline was designed and structured based on three factors: primary feeling within the space, the function of space, and capacity/number of space users. Figure 2 presents the relationship between provided virtual spaces and their designated guidelines.

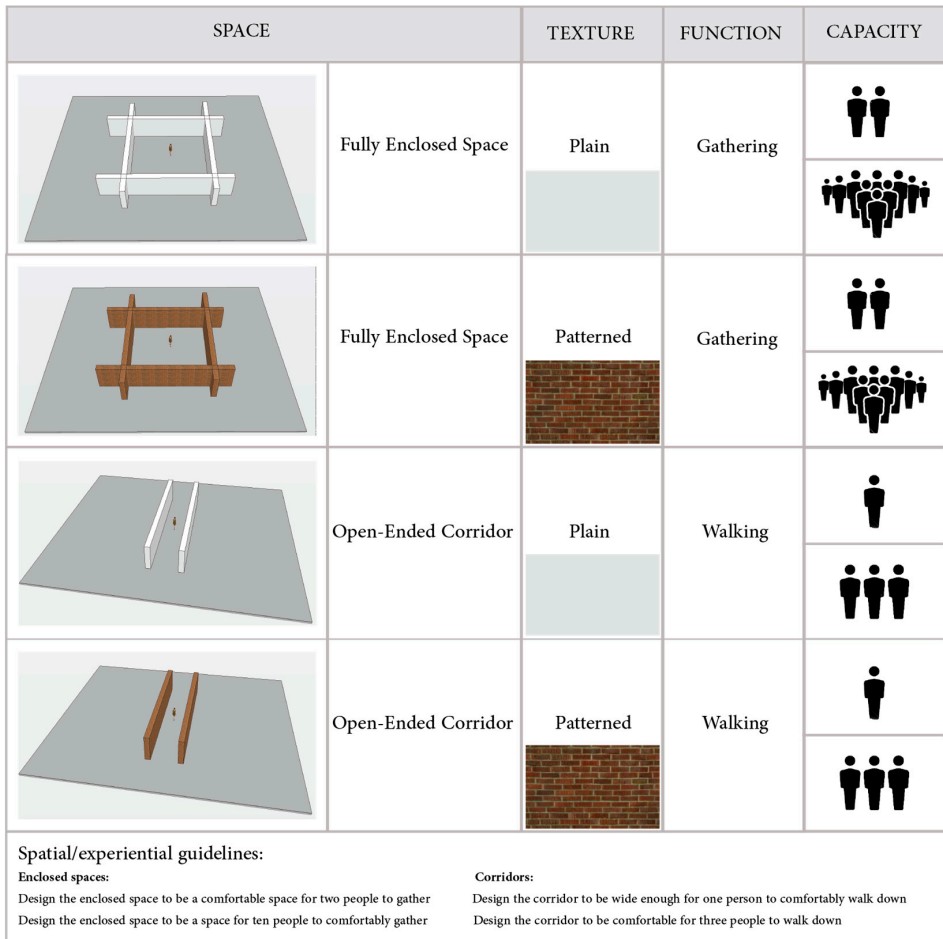

**Figure 2.** Developed models and adapted spatial/experiential guidelines for each space.

In step 3, a pilot was conducted to test the quality of the experimental setup and estimate the required average time that participants needed to complete the experiment using both systems. Each participant in the pilot worked on two virtual models (an enclosed space and a corridor space) using a guideline for each, once in the DT system and then IVRIE. Four .skp files were saved as design results of each participant, along with a questionnaire comprising five questions answered by the participant regarding their background. Based on the gathered information, the understandability of experiential/spatial guidelines and participants' possible questions about each guideline was studied and reviewed. In addition, the overall type and format of the PPQ questions were tested and finalized. The final version of PPQ consisted of nine objective questions with the focus on participants' background, such as age, gender, and major, along with their levels of familiarity with 3D models and experience in designing in IVR systems. Each question in PPQ is related to a factor that may be influential in affecting participants' spatial perception and cognition in virtual environments. These factors were utilized as perception filters in classifying the study's sample demographic profile. The PPQ can be found in the Appendix A section of this paper. For this study, the sample consisted of 60 participants comprising design students and professionals in landscape architecture, architecture, industrial design, art + design, and graphic design, as well as design faculty, engineering students, and professionals in civil engineering, chemical engineering, computer science, and other fields. Table 1 presents the sample population, categorized by perception filters.

**Table 1.** Sample demographics.

| Perception Filter | Sub-Groups | Population Percentage |
|---|---|---|
| Age range | 18–25 | 40% |
| | 26–35 | 44% |
| | 36–45 | 8% |
| | >46 | 8% |
| Gender | Male | 48% |
| | Female | 52% |
| Educational level | Bachelor's degree | 53% |
| | Master's degree | 27% |
| | Ph.D. | 12% |
| | Other | 8% |
| Major | Landscape architecture | 38% |
| | Architecture | 34% |
| | Other | 28% |
| Professional design experience | With | 57% |
| | Without | 43% |
| Years of professional design experience | <2 | 24% |
| | 2–5 | 50% |
| | 6–10 | 6% |
| | >10 | 20% |
| Level of familiarity with 3D environments in DT | Very | 48% |
| | Somewhat | 25% |
| | Not very | 17% |
| | Not at all | 10% |
| Level of familiarity with 3D immersive VR environments | Very | 18% |
| | Somewhat | 27% |
| | Not very | 23% |
| | Not at all | 32% |
| Level of direct design experience with IVRIE | A great deal | 8% |
| | Some | 12% |
| | A little | 12% |
| | None | 68% |

Step 4 included data collection from both data sources; design results of participants utilizing IVRIE and DT systems as .skp files; and their answers to nine questions on PPQ. In step 5, data refinement, statistical analyses, and the interpretation of final results were conducted. Figure 3 presents the sequence of steps in the experiment plan.

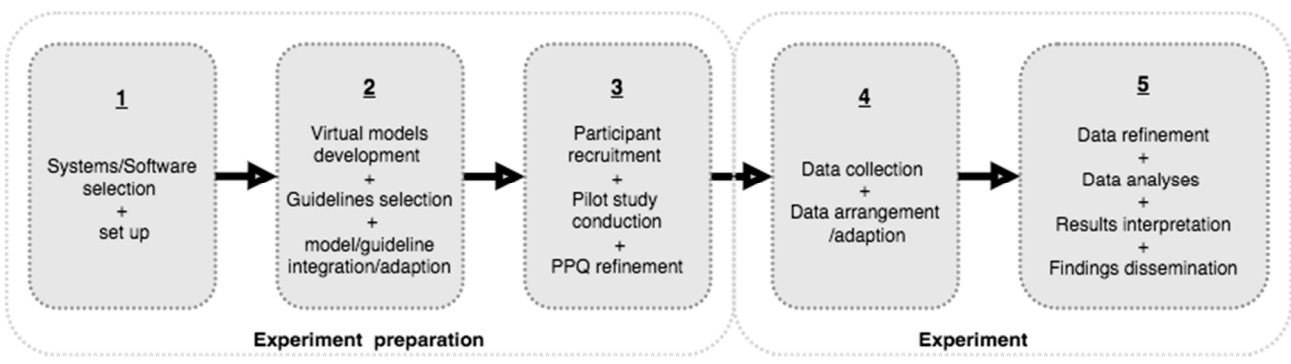

**Figure 3.** Flowchart of the experiment design.

## 4. Results

Data collection for this study was performed by running experiments. The experiments were conducted one participant at a time. The data from each participant were collected from two sources: the measurements of design results in each system and the answers to questions on PPQ. The extracted data from the measurements comprised 16 numerical values: the product of each participant's redesigned eight spaces in the DT system and eight in IVRIE. The eight numerical values gathered from each system consisted of four measurements of the enclosure spaces' inner area in square feet, and the four measurements of the width of the corridor spaces in feet. After each participant completed all the design tasks using both systems, their results were saved as '.skp' files. The total number of questions on PPQ was nine, and participants were asked to answer all questions that matched their profile. The one optional question was the number of years of professional design experience (see questions 5 and 6 on PPQ in Appendix A section). Each question on the PPQ represented a factor that it was felt may influence users' spatial perception and decision-making and was used as a perception filter in this study. The extracted quantitative data from the measurements of spaces in both DT and IVRIE systems were refined and tested through interquartile ranges to identify possible outliers before being used for statistical tests. In descriptive statistics, the interquartile range (IQR) is a measure of statistical dispersion, which shows the spread of the data and identifies any value that lies in an abnormal distance from other values in a random sample from a population.

The statistical tests compared the numerical values between the two systems; therefore, if a space size as a numerical value was an outlier in one system (i.e., in the DT system), it was removed from the data pool along with its paired numerical value from the other system (i.e., IVRIE). Additionally, if participants had three or more outliers in any system (3 or more outlier values between the total number of 16 numerical values collected for each participant), they and all their data were removed from the sample population and data pool. Based on the data refinement protocol of this study, six participants were removed from the sample demographic because of the excess in the number of their outliers.

At the end of the data refinement process, the refined data of space size measurements in each system were classified based on the sequence of questions on the PPQ. Each question on the PPQ is categorized as a perception filter. The answer options to each question were used as the categorizer and divider of the sample population into sub-groups based on the number of participants who selected the same answer options. Based on this sample population classification, the population percentage of each sub-group was calculated. Then, for the population of each sub-group, the design results' measurements were extracted from the data pool and analyzed. Data analyses consisted of performing various two-sample *t*-tests and calculating the *p*-values to identify significant levels of space size variations between the two systems. The *p*-value is an indicator of the statistical evaluation of two populations' averages. The *p*-values in the analyses of this study indicate the significant level of the average size differences for each space when designed by participants once in IVRIE and once in the DT system (i.e., the mean of the size of all corridors for three people walking, designed in a DT system compared with the mean of the same corridors, designed in IVRIE by all participants). The accuracy of statistical comparisons by two-sample *t*-tests could be higher or lower based on the designated significance level for the *p*-value. The level of significance adopted for all statistical comparisons in this study was 0.05, which generates a 95% confidence interval for determining the significance level of the differences in the average size of similar spaces designed by participants in both systems. Based on the statistical inferential of *p*-value = 5%, any calculated *p*-value lower than 0.05 was an identifier of a significant difference in the average size of each space designed in IVRIE with its paired space designed in the DT system.

The results and analyses of this study were categorized based on the proposed perception filters and their division of the sample population into sub-groups. Sub-groups are specific categories for each perception filter reflecting participant backgrounds and characteristics. In this study, nine perception filters were tested and analyzed as factors that

could impact spatial perception variations and users' spatial decision-making between the DT system and IVRIE. The perception filters are as follows: (1) age, (2) gender, (3) education level, (4) major, (5) having professional design experience, (6) amount of professional design experience, (7) familiarity with 3D environments in a DT system, (8) familiarity with 3D VR environments, and (9) having direct design experience in IVRIE. Based on the division of the sample population into different groups regarding the classifications of the perception filter, the design results of each group were compared between the two systems. The utilized criterion for comparing the variations in user perception between the two systems was the differences in the size of all designed spaces in DT and IVRIE systems by the participants in each sub-group. The results of statistical analyses of perception filters are presented subsequently.

*4.1. Age Range*

This perception filter comprised four groups: "18–25", "26–35", "36–45", and "46 and more" years of age. The statistical analyses compared the sizes of corridors spaces designed in the two systems by participants in each 'age range' sub-group.

A significant statistical difference for all age ranges was shown in the average width of corridors designed for three people walking, either in plain or patterned textured corridors, and the average width of corridors designed for one person walking in patterned corridors. There were no significant differences in the average width of plain corridors.

The comparisons of the size of enclosure spaces revealed a significant statistical difference for all age range groups between the two systems for enclosed spaces for gathering ten people, either with plain or patterned textures. In addition, the average size of plain enclosed spaces for gathering two people showed significant differences when designed by participants in age ranges of "18 to 25" and "46 and older". None of the participants in any age range groups designed the patterned enclosed spaces for gathering two people significantly differently in size between the two systems. Table 2 summarizes statistical analyses of the "Age range" perception filter and its sub-groups (sample population classifications) regarding comparing design results produced by participants utilizing IVRIE and DT systems. Figure 4 presents the space significant/non-significant size variations based on the division of sample population in "Age range" perception filter.

**Table 2.** Age range perception filter and comparisons of space size variations between IVRIE and DT systems.

| Perception Filter | Sub-Groups | Space Type | Fully Enclosed Space | | | | Open-Ended Corridor | | | |
|---|---|---|---|---|---|---|---|---|---|---|
| | | Function | Gathering | | | | Walking | | | |
| | | Texture | Plain | | Patterned | | Plain | | Patterned | |
| | | Capacity | 2 | 10 | 2 | 10 | 1 | 3 | 1 | 3 |
| | | Population Percentage | *p*-Value | | | | | | | |
| Age Range | 18–25 | 40% | 0.03 * | 0.02 * | 0.6 | 0.00 * | 0.1 | 0.02 * | 0.01 * | 0.00 * |
| | 26–35 | 44% | 0.07 | 0.02 * | 0.4 | 0.00 * | 0.1 | 0.00 * | 0.00 * | 0.00 * |
| | 36–45 | 8% | 0.5 | 0.04 * | 0.5 | 0.00 * | 0.9 | 0.01 * | 0.01 * | 0.01 * |
| | >46 | 8% | 0.03 * | 0.02 * | 0.9 | 0.00 * | 0.1 | 0.01 * | 0.01 * | 0.01 * |

The (*) symbol indicates significant *p*-value.

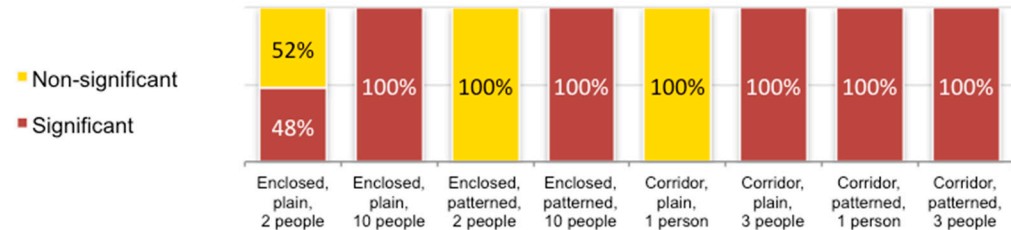

**Figure 4.** Percentage of participants with significant/non-significant size variations for each space in age range perception filter.

*4.2. Gender*

This perception filter consisted of "Males" and "Females" as subgroups. The comparisons of the average width of plain and patterned corridors for three people walking and patterned corridors for one person were significantly different for both genders between the two systems. The spatial decisions for both genders of the average width of plain corridors for one person walking were not different and resulted in non-significant size differences between the two systems. Between the two systems, the results of spatial decisions of both genders for enclosed spaces with a capacity for ten people, either with or without the presentation of texture, resulted in significant size differences, whereas patterned enclosed space for two people did not show any significant differences. The average size of plain enclosed spaces for two people was significantly different between the two systems when designed by male participants. In contrast, female participants' design results were not significantly different in the average size for these spaces. Table 3 summarizes statistical analyses of the "Gender" perception filter and its sub-groups regarding comparing design results produced by participants utilizing IVRIE and DT systems. Figure 5 presents each space significant/non-significant size variations based on the division of sample population in the "Gender" perception filter.

**Table 3.** Gender perception filter and comparisons of space size variations between IVRIE and DT systems.

| | | Space Type | Fully Enclosed Space | | | | Open-Ended Corridor | | | |
|---|---|---|---|---|---|---|---|---|---|---|
| | | Function | Gathering | | | | Walking | | | |
| | | Texture | Plain | | Patterned | | Plain | | Patterned | |
| | | Capacity | 2 | 10 | 2 | 10 | 1 | 3 | 1 | 3 |
| Perception Filter | Sub-Groups | Population Percentage | *p*-Value | | | | | | | |
| Gender | Male | 48% | 0.04 * | 0.02 * | 0.6 | 0.00 * | 0.1 | 0.01 * | 0.01 * | 0.00 * |
| | Female | 52% | 0.06 | 0.03 * | 0.7 | 0.00 * | 0.1 | 0.01 * | 0.00 * | 0.00 * |

The (*) symbol indicates significant *p*-value.

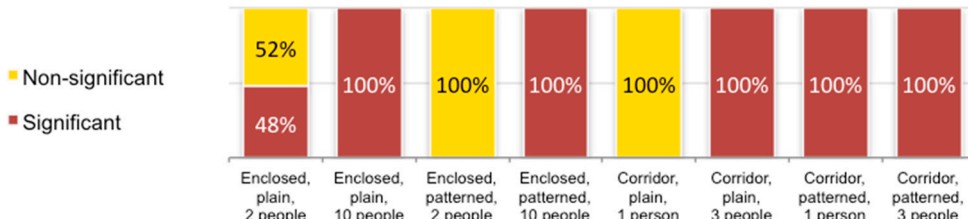

**Figure 5.** Percentage of participants with significant/non-significant size variations for each space in gender perception filter.

### 4.3. Educational Level

This perception filter comprised four subgroups: Bachelor's degree, Master's degree, Ph.D., and other. The "Other" subgroup, with 8% of the sample population, consisted of undergraduate students (juniors and seniors) in architectural and engineering majors.

The comparisons of design results of all educational levels between the two systems declare that patterned corridors, either for walking one or three people and patterned enclosed spaces for ten people, significantly differed in the average width and size between the two systems. The average size of the patterned enclosure spaces for gathering two people and plain corridors for one person did not show significant size variations between DT and IVRIE for any educational level sub-groups. The average size of enclosed spaces for gathering two or ten people was significantly different between the two systems when designed by participants with "Bachelor's degree" and "Other" (undergraduate), and not significantly different for Master's and Ph.D. educational levels. The average width of plain corridor spaces for three people was significantly different between the two systems designed by all participants in any education level except the "Other" (undergraduate) level. Table 4 summarizes statistical analyses of the "Educational level" perception filter and its sub-groups regarding comparing design results produced by participants utilizing IVRIE and DT systems. Figure 6 presents each space significant/non-significant size variations based on the division of sample population in "Educational level" perception filter.

**Table 4.** Educational level perception filter and comparisons of space size variations between IVRIE and DT systems.

| Perception Filter | Sub-Groups | Space Type | Fully Enclosed Space | | | | Open-Ended Corridor | | | |
|---|---|---|---|---|---|---|---|---|---|---|
| | | Function | Gathering | | | | Walking | | | |
| | | Texture | Plain | | Patterned | | Plain | | Patterned | |
| | | Capacity | 2 | 10 | 2 | 10 | 1 | 3 | 1 | 3 |
| | | Population Percentage | *p*-Value | | | | | | | |
| Educational level | Bachelor's | 53% | 0.04 * | 0.02 * | 0.6 | 0.00 * | 0.1 | 0.01 * | 0.01 * | 0.00 * |
| | Master's | 27% | 0.06 | 0.04 * | 0.5 | 0.00 * | 0.1 | 0.01 * | 0.00 * | 0.00 * |
| | Ph.D. | 12% | 0.04 * | 0.00 * | 0.9 | 0.00 * | 0.2 | 0.01 * | 0.04 * | 0.02 * |
| | Other | 8% | 0.1 | 0.06 | 0.9 | 0.01 * | 0.7 | 0.7 | 0.01 * | 0.03 * |

The (*) symbol indicates significant *p*-value.

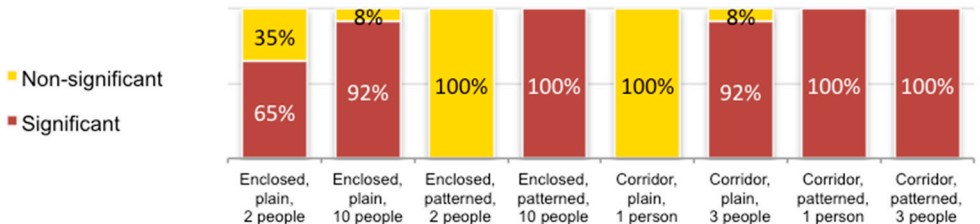

**Figure 6.** Percentage of participants with significant/non-significant size variations for each space in educational level perception filter.

### 4.4. Major

This perception filter comprised three subgroups: landscape architecture, architecture, and other. The "Other" major group included participants from industrial design, graphic design, art + design, civil engineering, chemical engineering, computer science, and other fields. The analyses indicate that between the two systems, the design results of all participants, regardless of their majors, were significantly different for patterned corridor spaces for one person walking and corridors for three people (both plain and patterned texture). In addition, the size of enclosure spaces for the gathering of ten people (both plain

and patterned texture) was significantly different. On the other hand, spatial decisions of participants in all "Major" sub-groups, between DT and IVRIE systems for plain corridor space for one person walking and enclosure spaces for gathering two people (either plain or patterned texture), were the same, and did not result in any significant differences in the average width or inner area of these space categories. Table 5 summarizes statistical analyses of the "Major" perception filter and its sub-groups regarding comparing design results produced by participants utilizing IVRIE and DT systems. Figure 7 presents each space significant/non-significant size variations based on the division of sample population in "Major" perception filter.

**Table 5.** Major perception filter and comparisons of space size variations between IVRIE and DT systems.

| | | Space Type | Fully Enclosed Space | | | | Open-Ended Corridor | | | |
|---|---|---|---|---|---|---|---|---|---|---|
| | | Function | Gathering | | | | Walking | | | |
| | | Texture | Plain | | Patterned | | Plain | | Patterned | |
| | | Capacity | 2 | 10 | 2 | 10 | 1 | 3 | 1 | 3 |
| Perception Filter | Sub-Groups | Population Percentage | *p*-Value | | | | | | | |
| Major | Landscape architecture | 38% | 0.09 | 0.02 * | 0.5 | 0.00 * | 0.1 | 0.01 * | 0.00 * | 0.00 * |
| | Architecture | 34% | 0.06 | 0.03 * | 0.5 | 0.00 * | 0.1 | 0.00 * | 0.00 * | 0.00 * |
| | Other | 28% | 0.05 | 0.04 * | 0.6 | 0.00 * | 0.1 | 0.01 * | 0.00 * | 0.00 * |

The (*) symbol indicates significant *p*-value.

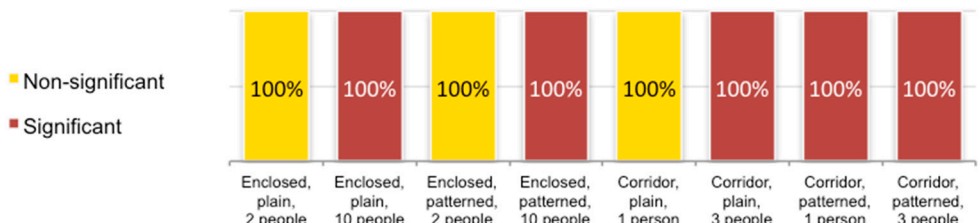

**Figure 7.** Percentage of participants with significant/non-significant size variations for each space in major perception filter.

### 4.5. Professional Design Experience

This perception filter consisted of two subgroups: "With" and "Without" design experience. The analyses found that the performance and design results for both groups utilizing both systems were similar. Both participants, with or without professional design experience, made different spatial decisions for corridors for three people (either with plain or patterned textures), patterned corridors for one person, and enclosure spaces for ten people gathering (plain and patterned texture) between the two systems. Thus, the average width of these corridors and the average size of enclosure spaces designated for ten users were significantly different. Conversely, the average size of enclosure spaces for gathering two people, either in plain or patterned texture, and the average width of plain corridors for one person did not show any significant differences between the two systems. Table 6 summarizes statistical analyses of the "Professional design experience" perception filter and its sub-groups regarding comparing design results produced by participants utilizing IVRIE and DT systems. Figure 8 presents each space significant/non-significant size variations based on the division of sample population in "Professional design experience" perception filter.

**Table 6.** Professional design experience perception filter and comparisons of space size variations between IVRIE and DT systems.

| Perception Filter | Sub-Groups | Space Type | Fully Enclosed Space | | | | Open-Ended Corridor | | | |
|---|---|---|---|---|---|---|---|---|---|---|
| | | Function | Gathering | | | | Walking | | | |
| | | Texture | Plain | | Patterned | | Plain | | Patterned | |
| | | Capacity | 2 | 10 | 2 | 10 | 1 | 3 | 1 | 3 |
| | | Population Percentage | *p*-Value | | | | | | | |
| Professional design experience | With | 57% | 0.07 | 0.02 * | 0.4 | 0.00 * | 0.1 | 0.00 * | 0.00 * | 0.00 * |
| | Without | 43% | 0.07 | 0.02 * | 0.6 | 0.00 * | 0.1 | 0.01 * | 0.00 * | 0.00 * |

The (*) symbol indicates significant *p*-value.

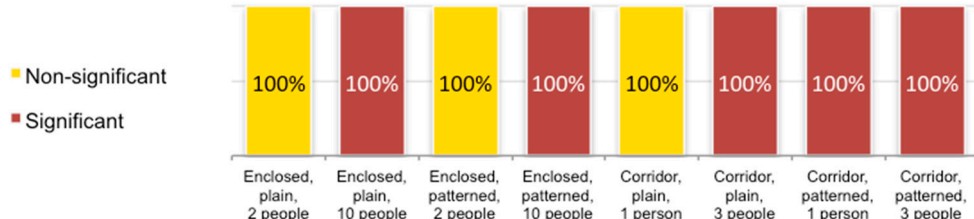

**Figure 8.** Percentage of participants with significant/non-significant size variations for each space in professional design experience perception filter.

### 4.6. Period of Professional Design Experience

This perception filter is a sub-branch of "Professional design experience" analyses and is the statistical analysis of 57% of the participants in the sample population who were identified "with" professional design experience. This perception filter comprised four subgroups "Fewer than 2", "2–5", "6–10", and "More than 10" years of professional experience.

The statistical analyses revealed that regardless of the period of professional design experience, the spatial decisions of participants for patterned corridors for one and three people walking, plain corridors for three people, and the patterned enclosure spaces for the gathering of ten people resulted in significant size differences for these spaces between the two systems. Conversely, the average size of the patterned enclosure spaces for gathering two people and the plain corridor for one person does not show significant size variations between the two systems for any participants in the sample. The average size of plain enclosure spaces for the gathering of two people was significantly different between the two systems when designed by the participants with "more than 10 years" of design experience. Other participants with fewer than 10 years of professional design experience did not design the space significantly different in size between the two systems. The spatial decision of participants in the sub-groups of "2–5" and "More than 10 years" in designing the plain enclosure spaces for the gathering of 10 people show significant size differences between the two systems. The other two groups with "Fewer than 2" and "6–10" years of experience designed the spaces with no significant difference in size between the two systems. Table 7 summarizes statistical analyses of the "Period of professional design experience" perception filter and its sub-groups regarding comparing design results produced by participants utilizing IVRIE and DT systems. Figure 9 presents each space significant/non-significant size variations based on the division of sample population in the "Period of professional design experience" perception filter.

**Table 7.** Period of professional design experience perception filter and comparisons of space size variations between IVRIE and DT systems.

| | | Space Type | Fully Enclosed Space | | | | Open-Ended Corridor | | | |
|---|---|---|---|---|---|---|---|---|---|---|
| | | Function | Gathering | | | | Walking | | | |
| | | Texture | Plain | | Patterned | | Plain | | Patterned | |
| | | Capacity | 2 | 10 | 2 | 10 | 1 | 3 | 1 | 3 |
| Perception Filter | Sub-Groups | Population Percentage | *p*-Value | | | | | | | |
| Years of professional design experience | <2 | 24% | 0.1 | 0.06 | 0.7 | 0.00 * | 0.3 | 0.01 * | 0.00 * | 0.00 * |
| | 2–5 | 50% | 0.07 | 0.02 * | 0.4 | 0.00 * | 0.1 | 0.00 * | 0.00 * | 0.00 * |
| | 6–10 | 6% | 0.4 | 0.4 | 0.09 | 0.00 * | 0.4 | 0.04 * | 0.00 * | 0.00 * |
| | >10 | 20% | 0.03 * | 0.02 * | 0.9 | 0.00 * | 0.1 | 0.01 * | 0.01 * | 0.01 * |

The (*) symbol indicates significant *p*-value.

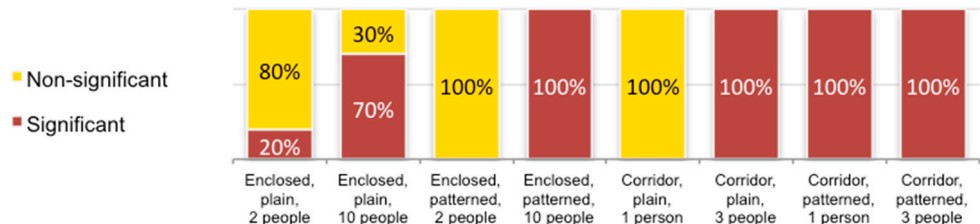

**Figure 9.** Percentage of participants with significant/non-significant size variations for each space in period of professional design experience perception filter.

### 4.7. Familiarity with 3D Environments in DT System

This perception filter comprised four subgroups: "Very familiar", "Somewhat familiar", "Not very familiar", and "Not at all familiar".

The analyses indicated that the spatial decisions of participants in all levels of "familiarity with 3D environments in DT system" were different between the two systems and resulted in significant differences in the average width of patterned corridors, either for walking one or three people and the average size of patterned enclosed spaces for ten people. Conversely, none of the participants in any levels of "Familiarity with 3D environments in DT system" designed patterned enclosed spaces for two people and plain corridors for one person significantly different in size and width between the two systems. Except for the participants in the "Not very familiar" level, other groups designed the plain corridors for three people with a significantly different average width between the two systems. The average size of plain enclosed spaces for ten people was significantly different between the two systems when designed by participants in "Very" and "Somewhat" levels of familiarity with 3D environments in DT. The average size of plain enclosed spaces for two people was significantly different between the two systems when designed by participants in the "Somewhat" familiarity level group. It did not show any significant size variations when designed by other groups of familiarity levels. Table 8 summarizes statistical analyses of the "Familiarity with 3D environments in DT system" perception filter and its sub-groups regarding comparing design results produced by participants utilizing IVRIE and DT systems. Figure 10 presents each space significant/non-significant size variations based on the division of sample population in "Familiarity with 3D environments in DT system" perception filter.

**Table 8.** Familiarity with 3D environments in the DT system perception filter and comparisons of space size variations between IVRIE and DT systems.

| | | Space Type | Fully Enclosed Space | | | | Open-Ended Corridor | | | |
| | | Function | Gathering | | | | Walking | | | |
| | | Texture | Plain | | Patterned | | Plain | | Patterned | |
| | | Capacity | 2 | 10 | 2 | 10 | 1 | 3 | 1 | 3 |
| Perception Filter | Sub-Groups | Population Percentage | *p*-Value | | | | | | | |
| Level of familiarity with 3D environments in DT | Very | 48% | 0.07 | 0.02 * | 0.4 | 0.00 * | 0.1 | 0.00 * | 0.00 * | 0.00 * |
| | Somewhat | 25% | 0.03 * | 0.01 * | 0.9 | 0.00 * | 0.1 | 0.02 * | 0.01 * | 0.01 * |
| | Not very | 17% | 0.1 | 0.06 | 0.9 | 0.01 * | 0.7 | 0.07 | 0.01 * | 0.03 * |
| | Not at all | 10% | 0.1 | 0.08 | 0.6 | 0.00 * | 0.5 | 0.00 * | 0.00 * | 0.00 * |

The (*) symbol indicates significant *p*-value.

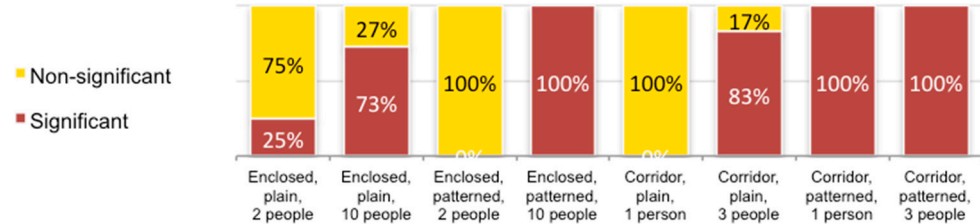

**Figure 10.** Percentage of participants with significant/non-significant size variations for each space in familiarity with 3D environments in DT system perception filter.

### 4.8. Familiarity with 3D Immersive Virtual Reality Environments

This perception filter comprised four subgroups: "Very familiar", "Somewhat familiar", "Not very familiar", and "Not at all familiar".

The statistical analyses revealed that in any levels of "Familiarity with 3D immersive VR environments", the spatial decisions of participants in designing the corridors for two people and enclosure spaces for ten people, either in plain or patterned texture, were different between the two systems and resulted in significant differences in the average width and size of these spaces. Conversely, none of the participants in any levels of "Familiarity with 3D immersive VR environments" designed patterned enclosed spaces for two people and plain corridors for one person significantly differently in size and width between the two systems. Except for the participants in the "Not very familiar" level, other groups designed the patterned corridor for one person significantly differently in average width between the two systems. The average size of plain enclosed space for two people was significantly different between the two systems when designed by participants in the "Not very" familiarity level. It did not show any significant size variations when designed by other groups of familiarity levels. Table 9 summarizes statistical analyses of the "Familiarity with 3D immersive Virtual Reality environments" perception filter and its sub-groups regarding comparing design results produced by participants utilizing IVRIE and DT systems. Figure 11 presents each space significant/non-significant size variations based on the division of sample population in "Familiarity with 3D immersive Virtual Reality environments" perception filter.

**Table 9.** Familiarity with 3D immersive Virtual Reality environments perception filter and comparisons of space size variations between IVRIE and DT systems.

| | | Space Type | Fully Enclosed Space | | | | Open-Ended Corridor | | | |
| | | Function | Gathering | | | | Walking | | | |
| | | Texture | Plain | | Patterned | | Plain | | Patterned | |
| | | Capacity | 2 | 10 | 2 | 10 | 1 | 3 | 1 | 3 |
| Perception Filter | Sub-Groups | Population Percentage | *p*-Value | | | | | | | |
| Level of familiarity with 3D immersive VR environments | Very | 18% | 0.05 | 0.00 * | 0.8 | 0.00 * | 0.2 | 0.03 * | 0.03 * | 0.02 * |
| | Somewhat | 27% | 0.07 | 0.02 * | 0.4 | 0.00 * | 0.1 | 0.00 * | 0.00 * | 0.00 * |
| | Not very | 23% | 0.02 * | 0.00 * | 0.5 | 0.00 * | 0.1 | 0.02 * | 0.1 | 0.04 * |
| | Not at all | 32% | 0.07 | 0.02 * | 0.6 | 0.00 * | 0.1 | 0.01 * | 0.00 * | 0.00 * |

The (*) symbol indicates significant *p*-value.

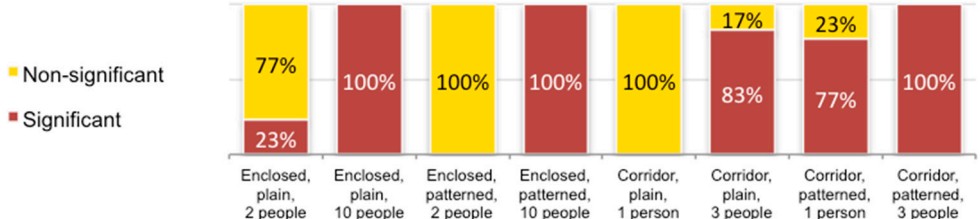

**Figure 11.** Percentage of participants with significant/non-significant size variations for each space in the familiarity with 3D immersive Virtual Reality environments perception filter.

### 4.9. Direct Design Experience with IVRIE

This perception filter comprised four sub-groups: "A great deal", "Some", "A little", and "None".

The analyses indicated that all participants, in any level of having "Direct design experience with IVRIE", made different spatial decisions between the systems for patterned corridors for one and three people, either with a plain or patterned texture, plain corridors for three people and patterned enclosed space for ten people. Conversely, none of the participants in any level of having "Direct design experience with IVRIE" designed plain corridors for one person and patterned enclosed spaces for two people significantly differently in width and size between the two systems. There were significant size differences for plain enclosed spaces for two people between the two systems when designed by participants in the "Some" level of having direct design experience with IVRIE. Additionally, participants with "Some" and "None" levels of having direct design experience with IVRIE designed plain enclosed spaces for ten people significantly differently in size between the two systems, whereas participants in the other levels did not make different spatial decisions for the size of these spaces differently between the two systems. Table 2 summarizes statistical analyses of perception filters and their sub-groups (sample population classifications) regarding comparing design results produced by participants utilizing IVRIE and DT systems. Table 10 summarizes statistical analyses of the "Direct design experience with IVRIE" perception filter and its sub-groups regarding comparing design results produced by participants utilizing IVRIE and DT systems. Figure 12 presents each space significant/non-significant size variations based on the division of sample population in the "Direct design experience with IVRIE" perception filter.

**Table 10.** Direct design experience with the IVRIE perception filter and comparisons of space size variations between IVRIE and DT systems.

| Perception Filter | Sub-Groups | Space Type | Fully Enclosed Space | | | | Open-Ended Corridor | | | |
| | | Function | Gathering | | | | Walking | | | |
| | | Texture | Plain | | Patterned | | Plain | | Patterned | |
| | | Capacity | 2 | 10 | 2 | 10 | 1 | 3 | 1 | 3 |
| | | Population Percentage | *p*-Value | | | | | | | |
| Level of direct design experience with IVRIE | A great deal | 8% | 0.08 | 0.1 | 0.5 | 0.00 * | 0.3 | 0.01 * | 0.01 * | 0.01 * |
| | Some | 12% | 0.03 * | 0.03 * | 0.6 | 0.00 * | 0.1 | 0.01 * | 0.01 * | 0.00 * |
| | A little | 12% | 0.1 | 0.05 | 0.8 | 0.01 * | 0.5 | 0.03 * | 0.00 * | 0.00 * |
| | None | 68% | 0.08 | 0.02 * | 0.5 | 0.00 * | 0.1 | 0.01 * | 0.00 * | 0.00 * |

The (*) symbol indicates significant *p*-value.

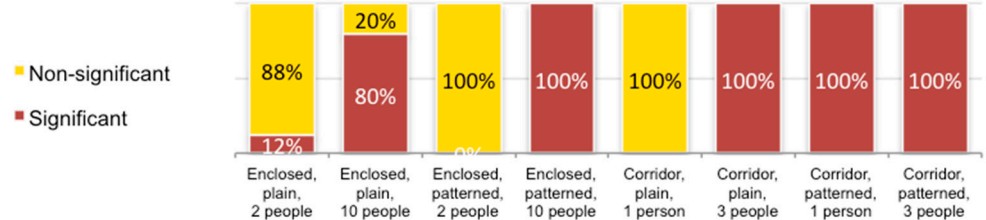

**Figure 12.** Percentage of participants with significant/non-significant size variations for each space in direct design experience with IVRIE perception filter.

*4.10. Population Percentage with Significant Space Size Variations between the Systems Primarily Due to the Impact of Perception Filters*

This branch of analysis indicated the percentage of participants with significant space size variations for each space between the two systems primarily due to the impact of perception filters. The results of statistical analyses are presented subsequently.

### 4.10.1. Age Range

This perception filter had an active role in affecting the spatial decision of 48% of participants to design the plain enclosed spaces for two people with significant size variations between the two systems. This perception filter did not impact the variation in design results of other spaces based on participants' age range groups. All the participants designed patterned and plain corridors for three people, patterned corridors for one person, and both plain and patterned enclosed spaces for ten people, significantly differently in size and width between the two systems. Additionally, none of the participants designed patterned enclosed spaces for two people and plain corridors for one person significantly differently between the two systems.

### 4.10.2. Gender

This perception filter had an active role in affecting the spatial decision of male participants (48% of the sample population) in designing the plain enclosed spaces for two people with significant size variations between the two systems. This perception filter did not have any impact on the variation in design results of other spaces based on participants' gender. All participants designed patterned and plain corridors for three people, patterned corridors for one person, and both plain and patterned enclosed spaces for ten people, significantly differently in size and width between the two systems. Additionally, none of the participants designed patterned enclosed spaces for two people and plain corridors for one person significantly differently between the two systems.

### 4.10.3. Educational Level

This perception filter had an active role in affecting the spatial decision of 65% of participants to design the plain enclosed spaces for two people with significant size variations between the two systems. Additionally, this perception filter affected the spatial decision of 92% of participants in designing the plain enclosed spaces for ten people and plain corridor spaces for three people, with significant differences in average size and width between the two systems. This perception filter did not impact the variation in other spaces' design results based on participants' educational levels. All participants designed patterned corridors for one and three people and patterned enclosed spaces for ten people significantly differently in size and width between the two systems. Additionally, none of the participants designed patterned enclosed spaces for two people and plain corridors for one person significantly differently between the two systems.

### 4.10.4. Major

This perception filter did not have any active role in specifying any proportion of the sample population in making different spatial decisions between the two systems because of their major. All the participants, regardless of their major, designed patterned corridors for one and three people, plain corridors for three people, and either plain or patterned enclosed spaces for ten people significantly differently in average width and size between the two systems. Additionally, none of the participants designed plain corridors for one person and plain or patterned enclosed spaces for two people, with significant differences in average width and size between the two systems.

### 4.10.5. Professional Design Experience

This perception filter did not have any active role in distinguishing the participant with or without professional design experience in making different spatial decisions between the two systems. All participants with or without "Professional design experience" designed patterned corridors for one and three people, plain corridors for three people, and either plain or patterned enclosed spaces for ten people significantly differently in the average width and size between the two systems. Additionally, none of the participants designed plain corridors for one person, and either plain or patterned enclosed spaces for two people significantly differently in average width and size between the two systems.

### 4.10.6. Familiarity with 3D Environments in DT System

This perception filter had an active role in affecting the spatial decision of 25% of participants in designing the plain enclosed spaces for two people and 73% of participants in designing the plain enclosed spaces for ten people, with significant size differences between the two systems. Additionally, this perception filter impacted the spatial decisions of 83% of participants in designing plain corridors for three people with significant average width differences between the two systems. This perception filter did not have any impact on the variation in design results of other spaces based on participants' familiarity level with 3D environments in the DT system. All the participants designed patterned corridors for one and three people and patterned enclosed spaces for ten people significantly differently in average width and size between the two systems. Additionally, none of the participants designed patterned enclosed spaces for two people and plain corridors for one person significantly differently between the two systems.

### 4.10.7. Familiarity with 3D Immersive Virtual Reality Environments

This perception filter had an active role in affecting the spatial decision of 23% of participants to design the plain enclosed spaces for two people with significant size variations between the two systems. Additionally, this perception filter affected the spatial decision of 77% of participants in designing the patterned corridor spaces for one person with significant differences in average width and width between the two systems. This perception filter did not impact the variation in other spaces' design results based on

participants' familiarity levels with 3D immersive VR environments. All the participants designed plain and patterned corridors for three people and plain and patterned enclosed spaces for ten people significantly differently in average width and size between the two systems. Additionally, none of the participants designed patterned enclosed spaces for two people and plain corridors for one person significantly differently between the two systems.

### 4.10.8. Direct Design Experience with IVRIE

This perception filter had an active role in affecting the spatial decision of 12% of participants in designing the plain enclosed spaces for two people and 80% of participants in designing the plain enclosed spaces for ten people, with significant size differences between the two systems. This perception filter did not impact the variation in other spaces' design results based on participants' levels of having direct design experience with IVRIE. All the participants designed patterned corridors for one and three people, plain corridors for three people, and patterned enclosed spaces for ten people significantly differently in average width and size between the two systems. Additionally, none of the participants designed patterned enclosed spaces for two people and plain corridors for one person significantly differently between the two systems. Table 11 presents the perception filters and participants' percentages with significant different design results between the two systems. Figure 13 presents the population percentage with significant/non-significant size variations for all the spaces between the systems.

**Table 11.** Perception filters and participants' percentage with significantly different design results between the two systems.

| Space Type | Fully Enclosed Space | | | | Open-Ended Corridor | | | |
|---|---|---|---|---|---|---|---|---|
| Texture | Plain | | Patterned | | Plain | | Patterned | |
| Capacity | 2 | 10 | 2 | 10 | 1 | 3 | 1 | 3 |
| Perception Filter | Participants' Percentage | | | | | | | |
| Age range | 48% * | 100% | 0% | 100% | 0% | 100% | 100% | 100% |
| Gender | 48% * | 100% | 0% | 100% | 0% | 100% | 100% | 100% |
| Educational level | 65% * | 92% * | 0% | 100% | 0% | 92% * | 100% | 100% |
| Major | 0% | 100% | 0% | 100% | 0% | 100% | 100% | 100% |
| Professional design experience | 0% | 100% | 0% | 100% | 0% | 100% | 100% | 100% |
| Familiarity with 3D environments in DT | 25% * | 73% * | 0% | 100% | 0% | 83% * | 100% | 100% |
| Familiarity with 3D IVR environments | 23% * | 100% | 0% | 100% | 0% | 100% | 77% * | 100% |
| Direct design experience with IVRIE | 12% * | 80% * | 0% | 100% | 0% | 100% | 100% | 100% |
| Average | 28% * | 93% * | 0% | 100% | 0% | 97% * | 97% * | 100% |

100%: participants' percentage with significant different design results between the systems. 0%: participants' percentage with no significant different design results between the systems. 0% < * < 100%: participants' percentage with significant different design results between the systems due primarily to impact of perception filter.

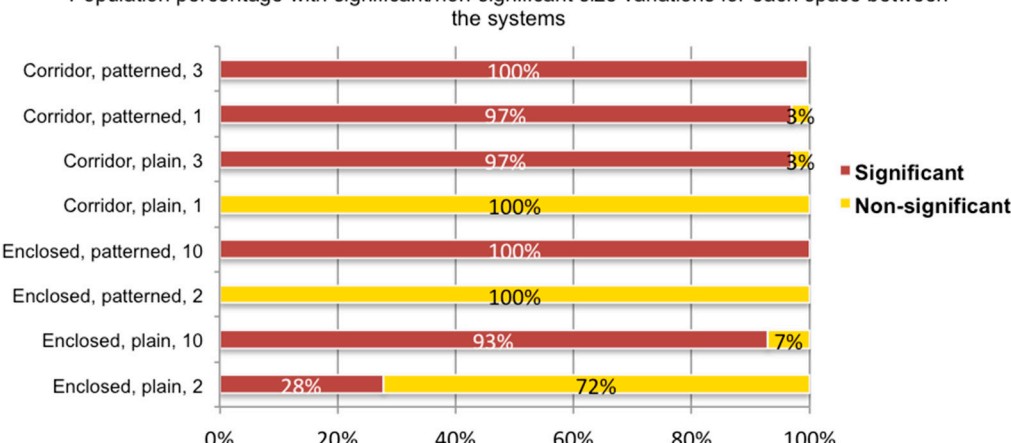

**Figure 13.** Population percentage with significant/non-significant space size variations between the systems.

### 4.11. Effective Perception Filters and Design Results' Size Variations between the Systems

The statistical analyses indicated that the impact levels of perception filters on participants' spatial decision-making between the IVRIE and DT systems are different and can result in both significant and non-significant size variations in designed spaces regarding the spatial function of a space, type of enclosure (fully enclosed space and open-ended corridor) and the presentation or absence of texture. Between the eight spaces designed by each participant once in IVRIE and once in the DT system, some spaces were designed with significant size variations by 100% of the participants. All the participants in the sample designed patterned enclosed spaces for ten people and patterned corridors for three people significantly differently in size and width between the two systems. Thus, there was no sign of any perception filters' impacts on distinguishing a portion of the sample population with different design outcomes for these spaces. Conversely, all the participants designed patterned enclosed spaces for two people and plain corridors for one person with non-significant size and width variations between the two systems. None of the perception filters caused differences in spatial decisions of the sample population for these two spaces. Most (97%) of the participants designed patterned corridors for one person with significant width differences between the two systems because of the impacts of "Familiarity with 3D IVR environments" perception filter. The plain corridors for three people were designed by 97% of participants with significant width differences between the two systems impacted by "Educational level" and "Familiarity with 3D environments in DT system" perception filters. Slightly fewer (93%) of the participants designed the plain corridor spaces for ten people with significant size differences between the two systems because of the impacts of "Educational level", "Familiarity with 3D environments in DT system" and "Direct design experience with IVRIE" perception filters. The average size of plain enclosed spaces for two people was significantly different between the two systems as the design results of 28% of the sample population because of the impacts of all perception filters except for the "Major" and "Professional design experience" perception filters. Table 12 presents the percentage of the population in producing significant and non-significant design results for each space and effective perception filters in the division of design results.

**Table 12.** Design results' size variations between the systems and effective perception filters.

| Space | Texture | Capacity | Design Results' Size Variations | | Effective Perception Filter |
|---|---|---|---|---|---|
| | | | Significant | Non-Significant | |
| Fully Enclosed Space | Plain | 2 | 28% | 72% | 1, 2, 3, 6, 7, 8 |
| | | 10 | 93% | 7% | 3, 6, 8 |
| | Patterned | 2 | 0% | 100% | - |
| | | 10 | 100% | 0% | - |
| Open-Ended Corridor | Plain | 1 | 0% | 100% | - |
| | | 3 | 97% | 3% | 3, 6 |
| | Patterned | 1 | 97% | 3% | 7 |
| | | 3 | 100% | 0% | - |

Perception Filters: 1, Age range; 2, Gender; 3, Educational level; 4, Major; 5, Professional design experience; 6, Familiarity with 3D environments in DT; 7, Familiarity with 3D IVR environments; 8, Direct design experience with IVRIE.

In this study, between the eight spaces utilized for testing the participants' spatial decision-making in designing them between IVRIE and DT systems, five spaces were significantly different in the average size or width between the two systems as the design results of 90% or more of sample population either being impacted by perception filters or not. The statistical analyses indicated that all these spaces, including plain and patterned corridors for walking three people, patterned corridors for walking one person, and patterned and plain enclosed spaces for gathering ten people, were designed on a smaller scale in the DT system compared with IVRIE. The average width of all corridors designed by participants in DT systems was narrower than their paired corridors designed in IVRIE. Additionally, the average size (inner area) of all enclosed spaces designed in DT was smaller than their paired spaces in IVRIE. Table 13 presents the means of width and size of designed corridors and enclosed spaces in each system.

**Table 13.** The average size of spaces designed in each system.

| Texture | Space | Function | Spatial Factor | Capacity | Mean | | *p*-Value |
|---|---|---|---|---|---|---|---|
| | | | | | IVRIE | DT | |
| Patterned | Open-Ended Corridor | Walking | Width (ft.) | 1 | 9 | 10.1 | 0.008 * |
| | Open-Ended Corridor | Walking | Width (ft.) | 3 | 18.7 | 21.2 | 0.001 * |
| | Fully Enclosed Space | Gathering | Area (ft$^2$) | 10 | 1046.9 | 1316.5 | 0.000 * |
| Plain | Open-Ended Corridor | Walking | Width (ft.) | 3 | 17.9 | 19.6 | 0.011 * |
| | Fully Enclosed Space | Gathering | Area (ft$^2$) | 10 | 1081.3 | 1241.4 | 0.023 * |

The (*) symbol indicates significant *p*-value.

## 5. Discussion

This study explored how the combination of user profile, characteristics of virtual spaces, and different features of IVRIE and DT systems impact users' spatial perception and decision-making and result in the production of different design results between the two systems. Two branches of quantitative data were utilized for the statistical analyses in this study. The first branch was extracted from the measurements of design results of the sample population for each system. These data consisted of information about the impacts of the characteristics of virtual spaces on users' spatial decision-making for each system, including textures, types of enclosure, and designated spatial functions. The second branch of data was extracted from PPQs, which collected participants' profile information. The combination of these two branches of data was used for the inferential statistical testing for this study.

This research was based on two main hypotheses. The first hypothesis assumed that some user characteristics would affect users' spatial perception in the IVRIE and DT systems and result in variations in spatial decision-making. The findings support this hypothesis: some users' spatial decision-making with regard to the sizes of similar spaces was different between the two systems from those of other users, due to variations in the particular characteristics. The second hypothesis assumed that differences in the level of presence and spatial perception between the IVRIE and DT systems would affect users' task-based design performance and spatial decision-making, resulting in different spatial design outcomes between the two systems. The results of the inferential statistical testing were not supportive of this hypothesis for all types of spaces. Although the levels of presence and spatial perception were different between the two systems, they did not always result in significant differences in spatial decision-making between the two systems. Based on the primary characteristics of the space (such as enclosure type, presentation of texture, and designated spatial function), users' spatial decisions were not different for all spaces between the two systems. In this study, the comparisons of participants' design results showed that all of the sample population designed some—but not all—spaces with significant average size variations between the two systems. The findings of this study can be summarized as follows:

- The spatial decisions for the scale of two types of spaces—patterned fully enclosed spaces for gathering ten people and pattern corridors for three people person walking—were different for all participant backgrounds, and resulted in significant size differences for these two types of spaces between IVRIE and DT systems.
- Three perception filters—"Educational level", "Familiarity with 3D environments in DT", and "Familiarity with 3D immersive Virtual Reality environments"—had active roles in impacting 97% of participants to design plain corridors for three people walking and patterned corridors for one person walking with significant differences in scale between the two systems.
- Three perception filters—"Educational level", "Familiarity with 3D environments in DT", and "Having direct design experience with IVRIE"—had active roles in impacting the spatial decisions of 97% of participants to design plain enclosed spaces with significant differences in scale between the two systems.
- Two perception filters—"Major" and "Professional design experience"—did not have any effective role in impacting the spatial decision of participants for the scale of any types of spaces between the IVRIE and DT systems and could be considered inactive perception filters.
- Overall, when using both IVRIE and DT systems for spatial design, two perception filters—"Educational level" and "Familiarity with 3D environments in DT"—played more active roles in impacting participants to design spaces with significant size variations between the two systems.

## 6. Conclusions and Future Vision

Although in recent years the amount of quantitative research focusing on the impacts of IVR on user perception and learning has increased and mostly concluded that virtual environments facilitate the process of learning architectural concepts [19,44–46], various factors and variables still need to be explored regarding the combination of the environment's features and user characteristics, as well as their impacts on users' spatial thinking and learning of spatial design. The main goal of the present research methodology was to integrate user-centered and system feature variables to better understand the impact on design outcomes when using IVRIE versus a DT system. In designing the experiment for this study, the authors assumed that because these two systems demonstrated critical differences in features such as sense of immersion (full vs. semi-immersion) and the way that users interact with design objects (direct vs. indirect), participants' spatial decisions would be fundamentally different, and the user background would represent an active variable enhancing such differences.

The similarity in spatial decisions for all participants between the two systems with regard to the width of a plain corridor accommodating one person could be due to a pre-framed spatial perception that humans have, based on their body size. Therefore, different virtual environments would have little impact. Additionally, the spatial decisions of all participants regarding the size of a patterned, fully enclosed space accommodating gatherings of ten people resulted in significantly smaller spaces when the design was developed in IVRIE as compared with the DT system. This may have been the consequence of the domination of spatial memory over the sense of immersion when estimating the logical space size for ten people in the DT system. In contrast, in the IVRIE, the sense of full immersion dominated spatial memory and may have led users to estimate the required space capacity based on their body scale, resulting in decisions leading to a smaller-sized space. In addition, the statistical analyses showed that when users made spatial decisions for two spaces of similar enclosure types and spatial functions, when a plain texture was replaced by a patterned texture, the probability of size variations significantly increased between the two systems. Between the two VR systems, the role of texture as a distractor or facilitator of users' spatial decisions, along with that of users' spatial memory and ability to engage in spatial thinking, require further exploration in future research. In future research, different textures, more complex virtual spaces in form and size, and different spatial/experiential guidelines will be tested. Additionally, in designing the spatial/experiential guidelines, a factor of spatial feeling will be added to the designated spatial function in each guideline, which makes the guidelines more spatially understandable for participants.

This research and its findings are a primary step toward understanding the factors impacting spatial design thinking and performance in IVR systems. In future research, various user-perception-related factors concerning spatial decision-making for spaces with more spatial complexity must be tested. Additionally, the correlations among design decisions based on sequential usage of virtual environments need further exploration.

**Author Contributions:** Conceptualization, S.A. and A.R.; methodology, S.A. and A.R.; validation, S.A.; formal analysis, S.A.; data curation, S.A.; visualization, S.A.; writing—original draft preparation, S.A.; writing—review and editing, S.A. and A.R. All authors have read and agreed to the published version of the manuscript.

**Funding:** This research received no external funding.

**Informed Consent Statement:** Informed consent was obtained from all subjects involved in the study.

**Data Availability Statement:** The data presented in this study are available on request from the corresponding author.

**Acknowledgments:** We thank all participants of this study for their generous participation.

**Conflicts of Interest:** The authors declare no conflict of interest.

## Abbreviations

| | |
|---|---|
| VR | Virtual Reality |
| IVR | Immersive Virtual Reality |
| IVRIE | Immersive Virtual Reality Interactive Environment |
| DT | Desktop system |
| 3D | Three-dimensional |

**Appendix A. Participant Profile Questionnaire**

Please answer the following questions:

1. Age:

　☐ 18-25　☐ 26-35　☐ 36-45　☐ 46-55　☐ +56　☐ Prefer not to respond

2. Gender:

　☐ Male　☐ Female　☐ Other　☐ Prefer not to respond

3. What is your highest educational degree earned?

　☐ Bachelor's degree　☐ Master's degree　☐ PhD　☐ Other

4. What is your major?

　☐ Landscape architecture　☐ Architecture　☐ Other

5. Do you have professional design experience?

　☐ Yes　☐ No

6. If you answered the previous question, yes, how many years of professional experience do you have?

　☐ Less than 2　☐ 2-5　☐ 6-10　☐ More than 10

7. How familiar are you with 3D environments in desktop system?

　☐ Very familiar　☐ Somewhat familiar　☐ Not very familiar　☐ Not at all familiar

8. How familiar are you with 3D immersive Virtual Reality environments?

　☐ Very familiar　☐ Somewhat familiar　☐ Not very familiar　☐ Not at all familiar

9. How much direct experience do you have in using a fully immersive Virtual Reality environment in design?

　☐ A great deal　☐ Some　☐ A little　☐ None

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
