# Peer review of "Scale Estimation for Design Decisions in Virtual Environments: Understanding the Impact of User Characteristics on Spatial Perception in Immersive Virtual Reality Systems"

_buildings, doi:10.3390/buildings12091461_

Round 1
Reviewer 1 Report
Please see the attached pdf file rich with comments and suggestions.

Author Response
Dear reviewer, thanks for your recommendations and comments. We have attempted to address all your concerns, comments, and questions in the attached pdf file. Your question regarding the reason for using the “desktop-based VR” word for semi-immersive environments could be answered as follows. All the virtual environments and 3D models observed through a screen (or the monitor of desktop systems) could be categorized as traditional VR or monitor-based VR. In these virtual environments, users have a sense of semi-immersion and feel partially immersed because of two well-known reasons. First, all the spatial data transferred to them are being received by passing a framed screen that acts as a divider (or obstacle) between the user and the virtual world. Second, when users are using desktop systems for browsing the virtual environment, the dominant sense of immersion, which is a full immersion, is shaped by the real world and their physical surroundings. Thus no user can feel equally about these two types of immersions simultaneously. That is why we abbreviated the word “desktop-based Virtual Reality” to DT system as the representer of seme-immersive virtual environments.
Many thanks for your time and consideration.
The Authors

Reviewer 2 Report
Dear Authors
The paper is interesting, and can be approved. However, it requires some reorganization. My suggestion is following:
The introduction chapter and the methodology chapter are both very long, and this makes it exhausting to the reader. You should create a new chapter for the background and transfer part of the text to the background.
Try to keep in the introduction the contextualization (3 paragraphs is enough), the Purpose statement, research questions, and the presentation of the structure of the manuscript. For the methodology, include only what refers to the research methods applied. Most of the text in the method section can be moved to the results.
The results chapter is also long, and can be synthesized.
The discussion chapter can be split into 2 parts, creating a final considerations chapter as a conclusion.
God job, go ahead
Author Response
Dear reviewer, thanks for your recommendations and comments. Based on your suggestions, the introduction section is divided into a new introduction and research background sections. Also, the new introduction section is revised based on your suggested format. The methodology section is revised, and all descriptions regarding the data types and statistical analyses are moved to the results section. The results section could not be shorter because of the consistency in presenting the findings; thus, many figures are added to the presented tables, making the whole section more understandable for readers. In the end, the discussion section is replaced by a new discussion and conclusion section based on your suggestion.
Many thanks for your time and consideration.
The Authors
Round 2
Reviewer 1 Report
I think that the reviews work!